# Anaerobic Digestion of Food Waste and Its Microbial Consortia: A Historical Review and Future Perspectives

**DOI:** 10.3390/ijerph19159519

**Published:** 2022-08-03

**Authors:** Shuijing Wang, Chenming Xu, Liyan Song, Jin Zhang

**Affiliations:** 1School of Resources and Environmental Engineering, Anhui University, Hefei 230039, China; x21301051@stu.ahu.edu.cn; 2College of Environment and Energy Engineering, Anhui Jianzhu University, Hefei 230601, China; chenmingxu95@163.com

**Keywords:** renewable energy source, food waste (FW), anaerobic digestion (AD), microbial consortia, integration of multi-omics, synthetic biology

## Abstract

Renewable energy source, such as food waste (FW), has drawn great attention globally due to the energy crisis and the environmental problem. Anaerobic digestion (AD) mediated by novel microbial consortia is widely used to convert FW to clean energy. Despite of the considerable progress on food waste and FWAD optimization condition in recent years, a comprehensive and predictive understanding of FWAD microbial consortia is absent and therefore represents a major research challenge in FWAD. The review begins with a global view on the FWAD status and is followed by an overview of the role of AD key conditions’ association with microbial community variation during the three main energy substances (hydrogen, organic acids, and methane) production by FWAD. The following topic is the historical understanding of the FWAD microorganism through the development of molecular biotechnology, from classic strain isolation to low-throughput sequencing technologies, to high-throughput sequencing technologies, and to the combination of high-throughput sequencing and isotope tracing. Finally, the integration of multi-omics for better understanding of the microbial community activity and the synthetic biology for the manipulation of the functioning microbial consortia during the FWAD process are proposed. Understanding microbial consortia in FWAD helps us to better manage the global renewable energy source.

## 1. Introduction

Food waste (FW) refers to the garbage generated during daily life by residents and various catering services. Generally, FW is classified as “avoidable” (leftovers and expired food) or “inevitable” (bones and eggshells) matters [1]. The Food and Agriculture Organization of the United Nations estimates that about 1.3 billion tons of food in the world becomes waste every year, accounting for about one-third of human food consumption. Figure 1 shows that FW accounts for 28–57% of domestic waste composition in different regions [2]. In 2015, China generated 186 million tons of municipal solid wastes (MSW), of which FW accounted for 37–62% [3]. The EU produces about 89 million tons of food waste every year [4]. The high water and organic matter content of FW makes it extremely perishable, and the deteriorated FW can facilitate the spread of microbial pathogens [5]. At the same time, the leachate produced by the accumulation of FW can cause groundwater pollution, and the methane produced by the degradation of food will hasten climate change if it is not treated appropriately [6].

To cope with the FW problem, various nations have been promoting a series of management strategies and developing FW treatment technologies, such as anaerobic digestion (AD). By 2015, there were more than 9000 relevant FW AD projects in operation in Germany, accounting for more than 80% of the biogas projects in Europe. In addition, there are approximately five million tons of fertilizer produced by FW every year [7,8]. The UK intends to increase the recycling rate of FW from current 10% to 70% by 2025 [9,10].

To date, there has been a lot of relevant research on the treatment of FW, from the initial physical methods, such as landfills and incineration, to aerobic composting and anaerobic digestion [11]. The landfill is a traditional treatment method, and its shortcomings are also evident. On the one hand, landfills take up a lot of land resources; on the other hand, landfills generate greenhouse gases and other harmful gases [12]. In recent years, as people have gradually realized that garbage has a strong potential for recycling and for power generation by incineration, composting and other methods have become increasingly popular [3,13]. Incineration is a relatively mature technology by which electricity can be generated through incineration of FW. However, many countries are reluctant to rely on incineration for dealing with FW due to the toxic and harmful chemicals, such as dioxin, produced by incineration [14]. Due to the large water content of FW (e.g., accounting for 80% of weight in China), the practical application of incineration is limited [15]. Although the operation of aerobic composting is simple, it has shortcomings that cannot be overlooked. First, in the process of composting, the leachate will pollute the environment. Second, nitrogen will be lost in the composting process, about 9.6–46% of which will be released into the atmosphere through NH_3_ [16,17]. AD is the most commonly used treatment method, as it has more obvious environmental benefits, such as renewable energy generation and waste reduction [13]. Under anaerobic conditions, the microorganism enriched by inoculation with sludge, poultry manure or by other resources converts FW to energy substances, such as organic acids, alcohol, hydrogen, and methane [18,19]. The fermentation broth also produced in the AD process contains rich N, P, and K elements, which can be used as fertilizer and soil conditioner to maintain the high yield of land and reduce the loss of nutrients in the environment, so as to realize the real waste reduction [20].

There has been a lot of research on the technical improvement of FWAD globally [21,22]. The essence of these technologies is manipulating the microbial consortia of AD process because the AD process is typical microbial mediation reaction. Although great progress has been made in recent years in studying the FWAD process’ key parameters and various approach strategies for enhancing FWAD performance, significant gaps still remain in our understanding of the microbial community structure and function during FWAD.

The objective of this paper is to review recent advances in our understanding of the microbial community during FWAD. First, this paper updates the current state of knowledge on global FWAD and discusses the key FWAD conditions (Parameters: temperature, pH, nutrients, and metal elements; Approaches: pretreatment and co-digestion) associated with microbial activity during FWAD generation of three main energy substances (hydrogen, organic acids, and methane). Then, the paper gives a historical overview of the FWAD microbial consortia based on the development of molecular biology technology. The combination of advanced high-throughput sequencing and isotope tracing used in FWAD microbial metabolism is also discussed. Finally, the integration of multi-omics for a better understanding of the microbial community activity and the synthetic biology for the manipulation of the functioning microbial consortia during the FWAD process are proposed and recommended.

## 2. The Current State of AD of FW

FW contains various organic macromolecules, such as proteins, fats, and sugars. In addition, FW also contains trace elements, such as Fe and Co, which facilitate the growth of microorganisms [23]. AD is a typical microbial-mediated process in which macromolecular substances in FW are transformed into hydrogen, organic acids, or methane through a series of reaction stages (these stages are not independent but rather cooperate to promote the anaerobic reaction) mediated by microorganisms, such as acidogenic microorganism and methanogenic microorganism. There are different classification systems for anaerobic reactions. The most accepted are the three-stage system (hydrolysis, acid production, and methane production) [24] and the four-stage system (hydrolysis, acidification, acid production, and methane production) [25] (Figure 2). The four-stage system further expands the acid production phase in the three-stage system into acidification and acid production.

Although AD can be identified as a three-stage and four-stage reaction, the degradation process essentially follows a similar pattern. First, there are numerous organic macromolecules in FW, and microorganisms often cannot directly use high molecular weight substances, so FW compounds need to be decomposed into smaller molecules through hydrolysis. The principal products in this stage are sugars, amino acids, and fatty acids. At the same time, the rate of hydrolysis is typically slow, so the hydrolysis stage is the rate-limiting step in AD [26]. Second, the products of the previous stage are further degraded into various volatile fatty acids (VFAs) and alcohol and lactic acid. Eventually, VFAs react to generate acetic acid, hydrogen, and carbon dioxide. Finally, methanogens can utilize the substances to generate methane. Since the AD process involves the division and cooperation of various microorganisms, the reaction is impacted by many parameters, such as temperature, pH, and the presence of nutrients and trace element (e.g., Fe and Co). Up to now, most of the existing technologies optimized the reaction conditions through physical, chemical, and biological control, so it is necessary to classify and integrate different influencing factors to further understand the mechanism of degradation by microorganisms (see Table 1). Hydrogen, organic acids, and methane are the three most important products during FWAD [27]. Hydrogen and methane are clean energy and organic acids and are important precursors of many industry chemicals. Usually, inhabitation methanogenesis would enhance the production of the hydrogen and organic acid [27]. Two-step AD, separating the hydrogen-producing microorganism and the hydrogen-consuming microorganism, yields more hydrogen than one-step AD [28]. In the following section, three main energy substances, hydrogen, organic acids, and methane, generation during FWAD and the associated key parameters are discussed.

### 2.1. Hydrogen Production

The process of hydrogen generation in AD mainly occurs in two stages. First, in the acidification stage, amino acids, fatty acids, and other substances produced after hydrolysis are converted by some acidifying bacteria into VFAs, such as acetic acid and propionic acid. At the same time, hydrogen, carbon dioxide, and other gases are released. According to the theory of Tanisho and Ishisho, the hydrogen production of AD is classified into two categories: pyruvate decarboxylation type and NAD+/NADH oxidation-reduction equilibrium regulation type [44]. The additional process occurs in the stage of hydrogen and acetic acid production, but because the hydrogen and acetic-acid-producing bacteria are easily affected by pH, the process is often accompanied by methane production.

#### 2.1.1. Temperature

To maintain the normal physiological activities of microorganisms, the environment must be in a relatively stable temperature range. When the hydrogen-producing microorganisms are in a frequently changing ambient temperature, their metabolism will be substantially affected. Temperature is a key parameter influencing hydrogen-producing microorganisms’ growth, physiological and metabolic activities. According to the most suitable temperatures, anaerobic microorganisms are mainly divided into thermophilic bacteria (about 55 °C) and mesophilic bacteria (about 35 °C), but for practical applications, most protocols use medium temperatures (about 37 °C) to produce hydrogen. Studies such as that of Kumar et al. indicate that the hydrogen production efficiency of the hydrogen-producing strain *Enterobacter cloacae* II T-BT08 increases with increasing temperature, reaching a maximum hydrogen production efficiency of 2.25 mol-H_2_/mol glucose at 36 °C [45]; Lin et al. found that when the temperature was between 15 and 34 °C, the hydrogen production rate increased with the increase in temperature [46]. Algapani et al. evaluated the effects of the thermophilic (55 °C) and hyperthermophilic (70 °C) condition on hydrogen production and found that the thermophilic condition is more beneficial for hydrogen production [47].

#### 2.1.2. pH

In addition to temperature, hydrogen-producing microorganisms are also very sensitive to pH. In a low-pH environment, it is often beneficial to inhibit the survival and reproduction of hydrogen-consuming microorganisms, thus improving the specific hydrogen production rate of the reaction system [48]. The optimal pH for maximum hydrogen production is found to be around 5–5.5 [28]. The change in pH value not only directly affects the growth state of the microbe but also causes changes in microbe morphology and structure [49] and ultimately leads to changes in the number and species of dominant microorganisms in the reaction system [50]. For instance, HS Shin et al. studied the effect of pH (4.5, 5.5, and 6.5) on hydrogen production, and the results showed that the maximum gas production was 70 mL/VS when the pH was 4.5. After that, with the increase in pH, the gas production gradually decreased, and the dominant hydrogen-producing microorganisms changed [51].

#### 2.1.3. Nutrient and Metal Elements

Anaerobic microorganisms need different nutrients to maintain their own metabolic activities. Among the nutrients, the C/N ratio and ammonia have been widely studied and are believed to be the key parameters. Metal elements, such as Fe, Co, and Ni, also play important roles. Kim D H et al. assessed the effect of adding sewage sludge (SWS) on the FW hydrogen production, and the results showed that when FW: SWS = 10:1, the hydrogen production increased by 13% compared with the FW treatment alone, and the reaction progress was significantly accelerated. This is because the sludge contains a variety of metal elements, such as Fe, which is conducive to the progress of the reaction [52]. In addition, research works have demonstrated that the performance and stability of AD are significantly related to the C/N ratio [53]. Anaerobic microorganisms require an environment with a relatively balanced nutritional structure, and a C/N ratio between 20 and 30 is thought to be optimal for anaerobic microorganisms [54]. Ammonia is also an essential nutrient for microorganisms. If the concentration of ammonia is low (50–200 mg/L), this will prohibit the AD of FW for biogas production, but if the concentration is too high (higher than 3 g/L), there will be a toxic inhibition effect on biogas production [55,56]. Pan et al. studied the effect of FW hydrogen production by adjusting the ammonia concentration (0–10 g/L). The results indicated that the addition of ammonia nitrogen can significantly improve hydrogen production when the total nitrogen concentration is less than 3.5 g/L [57].

#### 2.1.4. Pre-Treatment

A large number of degradable organic substances in FW exist as cells and micelles, and these structures are relatively stable, thus affecting the utilization of nutrients by microorganisms [58]. Therefore, the main purpose of pretreatment is to destroy the micelles and other structures by different means (chemical, biological, or physical) in order to increase the solubility and release the macromolecular substances into the surrounding environment. Therefore, microorganisms can more efficiently utilize nutrients for AD [58,59]. For example, hydrothermal pretreatment can effectively promote the dissolution of oils, sugars, and other macromolecular organics into small molecules to a certain extent, thereby improving the bioavailability of FW and regulating the composition of nutrients and making microorganisms better adapted to the surrounding environment. Li et al. showed that the hydrogen yield of FW after short-term wet heat pretreatment was significantly improved [60].

#### 2.1.5. Co-Digestion

Although FW can be used for AD alone, its characteristics of high salt, high oil levels, and high C/N ratio inhibit the activity of microorganisms [61]. For example, the optimal C/N ratio for microorganisms is generally between 15 and 20, while the C/N ratio of FW is usually higher. Higher C/N ratios cause the accumulation of organic acids, thus hindering the physiological activities of the microorganisms and finally leading to the termination of the reaction [54]. To regulate the nutritional structure and eliminate the inhibitory effect, various methods of combined fermentation can be employed. Accordingly, researchers have mixed fruit and vegetable waste, excess sludge, and other substances with FW to improve the stability of the reaction. When food waste and sludge are mixed in different proportions, the hydrogen output obtained is also different. A better effect is obtained when the mixing ratio is 1:1, and the hydrogen output increases from 36 mL/g-VS by FW along to 112 mL/g-VS by mixture [62]. Thus, mixed fermentation can adjust the C/N ratio to make the FW more suitable for the survival of microorganisms and thereby improve resource utilization.

### 2.2. Organic Acid

In addition to hydrogen, various organic acids are also essential products of FWAD; these include long-chain fatty acids, short-chain fatty acids, and lactic acid. For example, short-chain fatty acids can be used to produce biodegradable plastics [63] or as external carbon source for nitrogen and phosphorus removal in urban sewage [64], and lactic acid is an important raw material in the food and pharmaceutical industries. The production of organic acids mainly occurs in the stages of hydrolysis and acidification. First, insoluble organic compounds are hydrolyzed to long-chain fatty acids under the action of hydrolase and other microorganisms. Second, soluble organic compounds are further decomposed into short-chain fatty acids (acetic acid and propionic acid) under the action of acidification bacteria. At the same time, lactic acid and hydrogen are produced. In addition, VFAs are not only an essential product of AD, but they also are important indicators of the stability of the reaction. The failure of AD is usually due to the accumulation of numerous organic acids resulting in acid inhibition, which eventually leads to the loss of activity of anaerobic microorganisms.

#### 2.2.1. Temperature

Parallel to the process of hydrogen production, acidogenic bacteria can perform well at medium temperatures, and the appropriate temperature can promote the enzymatic reaction. For example, Komemoto K et al. studied the effect of temperature (25–65 °C) on the production of VFAs and biogas. The results showed that the total dissolution rate at high temperatures (55 and 65 °C) was significantly lower than that at medium temperatures, and the production efficiency of VFAs and biogas was also lower [65]. However, some studies have demonstrated that the yield of volatile acid can also be increased under ultra-high temperatures. For example, the mixture of FW and sludge was tested in 70 °C for acidification. The results showed that the acidification time was greatly shortened [66].

#### 2.2.2. pH

In the process of acid production, pH indirectly affects the concentration of volatile acids by affecting the activity of acid-producing microorganisms. Studies have shown that the production of volatile acids can be improved by promoting the acid-producing bacteria and reducing the activity of methanogens [67]. Generally, the activity of methanogens is higher when the pH is greater than 6.5 [68], so the setting of pH can affect the production of volatile acids [69]. However, this does not mean that lower pH will automatically increase the content of VFAs. Studies such as that of Wang have shown that when the pH is controlled at 4.0–6.0, as the pH decreases, the cumulative amount of VFAs also decreases [70]. Therefore, setting a reasonable pH range is a prerequisite for ensuring a smooth reaction. pH can also determine the type of volatile acid produced [50]. Owing to the different tolerances of various microorganisms to pH and the fact that fermentation types are determined by the dominant species of bacteria, different dominant bacteria will be produced under different pH, and then, different types of volatile acids will be produced.

#### 2.2.3. Nutrient and Metal Elements

FW methane production process often fails due to the accumulation of VFAs, and related studies have shown that some nutrients may cause organic acid accumulation. For instance, high concentrations of ammonia often cause the VFAs accumulation, but there is currently a dispute concerning the concentration range. Khanal et al. suggested that an ammonia concentration greater than 2000 mg/L would cause acid accumulation [71], while Borja et al. showed that an ammonia concentration greater than 5000 mg/L would cause significant VFAs accumulation [72]. In contract, some metal elements showed an advantage on VFAs prohibition effects. Studies have shown that adding a certain number of microelements (such as Co, Fe, and Cu) can not only delay the production time of VFAs but also significantly reduce the accumulation rate of VFAs. However, there was no difference under high temperature [73]. In addition, with the increase in carbon content, C/N increased from 35.9 to 40.4, while VFAs concentration decreased by 20.2% and 60.4% at the beginning and end of the reaction, respectively [74].

#### 2.2.4. Pre-Treatment

In addition to the pre-treatment methods listed in Table 1, there are additional distinct pre-treatments. For example, Song et al. investigated the effect on VFAs by setting the pH pre-treatment at different values. The results showed that the VFAs were reduced under different pH pre-treatments. The concentration and composition of VFAs have different effects. When the pH was 11.0, the concentration of VFAs was highest [75]. In addition, the combination of different pretreatment methods can also effectively increase the output of VFAs. For example, Elbeshbishy et al. combined ultrasonic and pH (acid and base) pre-treatment to obtain maximum VFAs output of 16,900 mg chemical oxygen demand (COD)/L under ultrasonic-acidic conditions [76]. Liu et al. concluded that ultrasonic basic pre-treatment increased the yield of VFAs by 68% compared with the untreated control [77].

#### 2.2.5. Co-Digestion

As mentioned above, the insoluble substances contained in FW limit the anaerobic fermentation process to a certain extent. The combination of diverse substances and FW can effectively regulate the nutritional structure of FW and enhance the stability of the reaction. Jia Lin et al. combined the fermentation of FW with fruit and vegetable waste according to 2:1 and 1:1 ratios; the results showed that the mixing ratio of 2:1 significantly increased the output of VFAs from 67 mg/L to 1100–1200 mg/L [78]. In addition, the combination of pre-treatment technology and fermentation can also play a role in adjusting the original conditions of FW, thereby promoting the conversion rate of VFAs. For example, Lim et al. pretreated the mixture of brown water and FW using microwaves. The results showed that microwave pre-treatment could increase the accumulation of VFAs, 21% higher than the control, and the treatment could also advance the transformation of other short-chain fatty acids to acetic acid [79].

### 2.3. Methane

The AD process produces biogas, a mixture of methane, carbon dioxide, hydrogen gas, and other gases. Among them, methane is one of the principal gases. Under strict anaerobic conditions, FW generates methane through the action of methanogens, and methanogens mainly generate methane through two ways: hydrogenotrophic methanogenesis [80] and acetoclastic methanogenesis [81].

#### 2.3.1. Temperature

Compared with hydrolytic bacteria and acidogenic bacteria, methanogens have a wider temperature range. However, the activity of methanogens is significantly inhibited when the temperature is lower than 20–25 °C or higher than 50 °C [82]. At present, the regulation of temperature in the methanogenesis process is generally divided into high temperature and medium temperature, in which high temperature can bring relatively higher methane production and substrate consumption rate [83]. However, this is controversial. According to the research of Komemoto K et al., under the high temperatures of 55 and 65 °C, biogas production efficiency was reduced, and the high temperature inhibited the activity of methanogenic microorganisms [65]. A study also indicated that the temperature significantly impacted the FWAD because it was easier to produce NH_4_+ under high temperatures, which inhibited the growth of methanogens [84].

#### 2.3.2. pH

Organic acids are formed in the process of producing acetic acid, and acetic acid is a substance that can be directly used by methanogens. However, many of the VFAs produced cannot be utilized by methanogens, resulting in the accumulation of VFAs and pH values below 5 [85]. If methanogens propagate in large numbers, the ammonia concentration will rise, and the pH will rise above 8. If the pH is already too high or too low, methanogens will be greatly affected. Because methanogens are more sensitive to the changes in pH than acidogenic bacteria [86], the suitable pH range of most anaerobic bacteria is 5–8.5, and the optimal pH of methanogens is 6.8–7.2. Therefore, when the accumulation of organic acids causes the decrease in pH, the consumption of H_2_ will be reduced, and the activity of methanogens will be far lower than that of acidogenic bacteria and will thus reduce their ability to consume organic acids, further leading to the accumulation of more organic acids and finally producing the phenomenon of “acidosis” to stop the reaction. Although the optimal pH range of methanogens is thought to be 6.8–7.2, some researchers believe that with different substrates and anaerobic technologies, the optimal pH will vary. Due to the relationship between pH and the concentrations of VFAs, bicarbonate, and CO_2_ in biogas, regulating VFAs concentration is particularly important for stabilizing pH [80].

#### 2.3.3. Nutrient and Metal Elements

There are many N-rich substances in the raw materials of FW, resulting in significant problems in the subsequent treatment process [87]. For example, Chen’s research demonstrated that when the ammonia concentration is greater than 2 g/L, the methanogenesis process is inhibited. Through microbial analysis, it has been shown that when the ammonia concentration is increased, microorganisms unrelated to the methanogenesis process are enriched, while *Methanobcterium* and *Methanopirillum* are inhibited [88]. Li et al. increased the carbon content in the reaction system by adding straw and then changed the C/N ratio. The results showed that methane production increased 41.3% when the C/N ratio increased from 20.3 to 28.5 [89]. Studies also demonstrated that Fe, Co, Ni, and other metal trace elements are also of great significance to methanogens. The addition of trace elements in the reaction system can prevent the accumulation of VFAs [90]. Metal elements also participate in the synthesis and activate many enzymes related to the methanogenesis process, and trace elements also improve the adaptability of methanogens to interference factors, such as Na+, in the environment. Richard Speece suggested that the importance of metal trace elements has been largely ignored and that this has hindered the development of anaerobic digestion methane production technology [91]. To date, the related studies of the effects of trace elements on AD performance are still limited.

#### 2.3.4. Pre-Treatment

Different pre-treatment approaches have different effects on the improvement of AD reaction efficiency. For example, Ma et al. evaluated the effects of pre-treatment methods (acid, thermal, thermo-acid, and pressure) on FW methane production. Cumulative gas production under pressure treatment was the highest, reaching 88 L/d, and the cumulative gas production under acid treatment was the lowest, reaching 1.8 L/d, indicating that all of the pre-treatment methods improved the AD capacity of FW to a certain extent [92]. In addition, some researchers have combined the pre-treatment technology with the fermentation technology to explore the impact on the methane production process. For example, FW combined with the residual sludge reduced the contents of total suspended solids (TSS), volatile suspended solids (VSS), and COD through heat treatment and ultrasonic pre-treatment. The results indicate that different pre-treatment technologies will have different effects and that such treatments can significantly improve methane production [93].

#### 2.3.5. Co-Digestion

The combined fermentation of FW and other substances can effectively improve methane production. Straw and animal manure produced in the production processes of agriculture and aquaculture are ideal combined fermentation substances. According to Li et al., FW has a low buffer capacity and is easy to acidify; corn straw cellulose cannot be directly used by anaerobic microorganisms and has low methane production; chicken manure has a low C/N ratio and low anaerobic digestion capacity, so the effect of single digestion using those three substances is not ideal, but the methane production can be greatly improved by mixing the three substances in a certain proportion [89]. Aragaw tried to enhance AD by adding rumen fluid based on the joint fermentation of FW and cow manure, and the results showed that the addition of rumen fluid enhanced the effect of joint fermentation by 24–47% relative to the control group [94]. For the agricultural residues, Ye et al. suggested that the combined fermentation with FW was more cost effective compared with the pretreatment, and the biogas production increased significantly after FW, pig manure, and straw were mixed in the correct proportions; at a ratio of 0.4:1.6:1, the biogas production was maximized, 71.6% higher than that using straw fermentation alone [89].

## 3. Studies of Anaerobic Microorganisms

Our understanding of AD microorganisms has been increasing with the development of molecular biotechnology. The initial research established the conditions suitable for microbial growth and focused on the isolation, purification, and cultivation of single microorganisms [95]. After that, molecular biology technology developed rapidly, and the research object was expanded from single strain to the entire microbial community using studies based on DNA and RNA by sequencing using low thorough sequencing techniques (DGGE, FISH, Clone library and TRFLP, examples please see Table 2) and high throughout sequencing techniques (454 and Illumina, examples please see Table 3). This greatly enhanced the study of the anaerobic reaction mechanism. 

### 3.1. Single Isolation Strains

Hydrogen-producing, acid-producing bacteria and methanogens are well characterized in the AD process. Hydrogen-producing bacteria can be subdivided into strict anaerobic bacteria and facultative anaerobic bacteria, the latter including *Clostridium* sp., *Enterobacterium* sp., and *Citrobacter* sp., among which *Clostridium* sp. and *Enterobacterium* sp. are the most studied. Obtaining efficient hydrogen-producing bacteria is an essential prerequisite for anaerobic digestion and hydrogen production from FW. For example, Oh Y K et al. isolated *Bacillus coagulans II* t-bt S1 from activated sludge. When glucose was taken as a substrate, the H_2_ production rate reached 2.49 mol H_2_/mol glucose [96].

This is because hydrogen-producing and acid-producing bacteria usually depend on each other in a mutualism, and the hydrogen-producing and acid-producing bacteria are also called “specialized mutual bacteria”. In 1967, Bryant et al. successfully isolated and purified the first hydrogen- and acid-producing bacteria [97], and more mutual hydrogen- and acid-producing bacteria have been found. Friedrich et al. found that some hydrogen-producing and acid-producing bacteria *Syntrophobolulus glycolicus* can co-culture with the methanogen *Spirillum henryi* to produce H_2_ and CO_2_ [98].

Methanogens are strictly anaerobic microorganisms. For food waste, there are rarely studies on methanogens isolation. The methanogens studied in food waste AD usually come from the inoculation source, such as activated sludge and others. Methanogens contain special enzymes to ensure the smooth progress of their own reactions; for example, the mcrA gene exists widely in methanogens [99]. Whether methane is produced by H_2_/CO_2_ or the acetic acid pathway, the mcrA gene product is required to catalyze the formation of methane [100]. At present, the method of identifying methanogens by recognizing the mcrA gene is widely used in the detection of the diversity of methanogens in the environment [101].

### 3.2. Application of DGGE Technology

In 1993, Muyzer et al. first used the denaturing gradient gel electrophoresis (DGGE) technology in the field of microbiology. By analyzing the genomic DNA of a sample, this technology was demonstrated to be useful in understanding the genetic characteristics of microbial communities [102]. Since then, the PCR-DGGE technology has been gradually applied to the study of anaerobic microorganisms, as these methods avoid the limitations of microbial culture and purification technology, thereby increasing our understanding of anaerobic reactions. Kim et al. found that hydrogen production can be enhanced after pretreatment for one day under pH 12.5, and through the DGGE analysis, it was found that alkaline pretreatment conditions can effectively prevent the transformation of microbial communities to non-hydrogen and acid-producing bacteria [103]. Xu et al. studied the effect of VFAs on the methanogenic community composition through DGGE under different organic load regulations. The results showed that the regulation of organic load could effectively promote methane production of FW within a certain range, and the methanogenic microbial community in FW would not change significantly. However, with the increase in organic load, the accumulation of acetic acid gradually leads to the destruction of the balance between acetoclastic methanogens and hydrogenotrophic methanogens, eventually leading to the reduction in methane production. At this time, some methanogens are inhibited, and *Methanosarcina mazi, Methanobcterium* sp., and *Methanocorpusculum* sp. eventually become the dominant species [104].

### 3.3. Application of FISH Technology

Although the DGGE technology has helped realize the transformation from the cultivation of single microorganisms to the research on the microbial community based on rRNA genes, there are still many drawbacks of this technology. For example, on the one hand, we cannot intuitively know the actual information concerning microbial morphology and spatial distribution [105]. On the other hand, we need to rely on PCR technology, which makes the operation process more complex and the accuracy slightly reduced. The emergence of fluorescence in situ hybridization (FISH) technology makes up for the shortcomings of the DGGE technology. First, the hybridization of fluorescent-labeled probes with nucleic acid sequences can be used to obtain pertinent morphological information. Second, FISH technology does not rely on PCR amplification. The operation process is further simplified, and the accuracy is improved; these advantages have led to FISH technology being widely used in the study of anaerobic microorganisms [106]. Chen et al. found that acid production from a combination of FW and activated sludge was significantly increased compared to when they were fermented separately. FISH technology revealed that the ratio of bacteria to archaea was the largest after combined fermentation at 4:1; the respective ratios of bacteria to archaea under the fermentation of FW and sludge alone were only 2.8:1 and 1.5:1, respectively, indicating that the increase in acid production was mainly due to the activity of anaerobic microorganisms and the proportions of bacteria and archaea [42]. Charles J. Banks and colleagues utilized the FISH technology to study the internal causes of the reaction failure caused by VFAs accumulation in the fermentation of FW with high-concentration ammonia loading and found that the fermentation was improved by adding trace elements. The results showed deficiency of selenium, essential for both propionate oxidation and syntrophic hydrogenotrophic methanogenesis leads to process failure. Only hydrogenotrophic methanogens were detected by the FISH technology in the samples with added trace elements [90].

### 3.4. Application of Cloning Library Technology

Similarly, the cloned library compensates for the shortcomings of the DGGE technology to some extent. In DGGE technology, it is difficult to analyze samples quantitatively, so cloned libraries can more accurately describe the characteristics of anaerobic microbial community changes. For example, Lotta Leven et al. studied the microbial community composition during the decomposition of domestic organic waste at 37 °C and 55 °C, and they employed a cloned library using 16 s rRNA specific bacteria and archaea. Cloning library analysis showed that the number of different clone sequences at medium temperatures was significantly higher than that at high temperatures, where *Bacteroidetes* (34% of the total clones) and *Chloroflexi* (27%) were the dominant genera, while the dominant genera at high temperature were *Thermomotogae* (61%), indicating that temperature had a significant effect on the microbial community during the methanogenesis process [107]. The CSTR reactor is used for anaerobic digestion, and the acid production and methane production are carried out in two steps, the so-called two-phase CSTR, which significantly improves gas production and stability compared to single-phase CSTR. J.W. Lim et al. analyzed the microbial communities of single-phase and two-phase CSTR reactors via clone library and FISH, respectively. The results showed that the differences might be explained by the differences in the microbial community structure [108].

### 3.5. Application of TRFLP Technology

Although DGGE and FISH technologies can remove the limitation of relying on microbial culture, the associated community fingerprints cannot be directly translated into taxonomic information [109]. Terminal restriction fragment length polymorphism (TRFLP) can reflect the population distribution in the microbial community and can explore the composition and quantity changes of the microbial community under different conditions [110]. Chen et al. used the TRFLP technology to assess the effects of different temperatures and the mixing ratio of FW and cow manure on the AD reaction. The analysis showed that archaea were mainly affected by temperature, and bacteria were mainly affected by temperature and FW/fecal ratio. The composition of the microbial community structure is greatly affected under different reaction conditions, which in turn affects the production of biogas. At 35 °C and the addition ratio of 90/10, the proportion of methane reaches the maximum of 65.2%. The diversity of the archaeal community under moderate temperature conditions is more abundant. As the temperature increases, the proportion of *Methanosarcina* increases from 70% to 90%, while *Methanobrevibacter* almost disappears [111]. Lucia Blasco et al. pretreated FW by autoclaving and analyzed the changes of the microbial community in the reactor under two different feed conditions, pretreated and untreated, through TRFLP. The results showed that although the same feed and reaction conditions were used, there were still different community structure characteristics between the control group and the pretreatment group, which explained the difference in methane production rates [112].

**Table 2 ijerph-19-09519-t002:** Bacterial and archaea taxonomic composition at phylum/order and genus/species level in anaerobic digestion reaction determination by low-throughout sequencing.

Feedstocks	Scale	Dominant Phylum (Bacteria)	Dominant Genus/Species (Bacteria)	Dominant Order (Archaea)	Dominant Genus/Species (Archaea)	Sequencing Platform	Ref.
FW + Anaerobic sludge	500 mL reactor	*ND*	*ND*	*ND*	*Methanosarcina mazei* *Methanobacterium* sp. *Methanoculleus marisniqri*	DGGE	[104]
FW+ brown water	5 L CSTR reactor	*Bacteroidetes Chloroflexi* *Proteobacteria Firmicutes*	*Lactobacillus* sp. *Acetobacter* *peroxydans* *Fusobacterium* sp.	*Methanosarcinales Methanomicrobiales*	*Methanoculleus* *Methanosarcina* *Methanosaeta*	FISH	[108]
Household waste	45 L reactor	*Bacteroidetes Chloroflexi* *Firmicutes* *Spirochaetes Thermotogae Actinobacteria Proteobacteria*	*ND*	*Thermoplasma Crenarchaeota* *Methanosarcinales Methanomicrobiales*	*Methanosarcina,* *Methanoculleus* *Methanobacterium.*	Cloning library analysis	[107]
FW+ Fresh cow manure	0.75 L CSTR reactor	*Bacteroidetes Firmicutes* *Proteobacteria Spirochaetes*	*Clostridia* *Bacteroidetes* *Petrimonas* *Bacteroides*	*ND*	*Methanosarcina* *Methanobrevibacter* *Methanobacterium Methanoculleus*	TRFLP	[111]
FW + sludge	11 L STRs reactors	*ND*	*ND*	*Methanosarcinales*	*Methanosarcina*	TRFLP	[112]

### 3.6. Application of High-Throughput Sequencing Technology

Although DGGE, FISH, TRFLP, and clone library played an important role in AD microbial consortia characterization, they could not provide the entire microbial community information. Starting with the 454 high-throughput sequencing and blooming by the second-generation sequencing launched by the Roche company, high-throughput sequencing was highly praised by *Nature* as the pioneering technology in this field [113]. Since then, Illumina Inc. and Applied Biosystems Inc. have launched Soxela and SOLiD technologies, respectively, which are jointly referred to as high-throughput sequencing technology. AD is a complex biological process involving many microorganisms, but technologies such as DGGE, FISH, and clone library significantly underestimate the community composition and only reflect a few dominant microorganisms, while high-throughput sequencing technology can analyze 100 million gene sequences at a time. The results can more accurately reflect the structural composition of the bacteria and archaeal communities under different classification levels according to research needs (Table 3). For example, Wu et al. studied the adaptability of microbial communities during thermophilic anaerobic treatment of FW using high-throughput sequencing to determine the sample community classification at different times. The results showed that there were 10 different “phylum” classifications and a small number of “other” classifications [114]. Dennehy et al. further reorganized the microbial community composition under different classification levels (phylum, family, and genus) during the joint reaction of FW and pig manure; their approach could more richly describe the structural changes of the microbial community during the AD [115]. Jing et al. used the high-throughput sequencing technology to study the effect of changes in total solids’ content on the efficiency of FW methane production. The results showed that as the total solids’ content increased, the relative abundance of *Chloroflexi* in bacteria decreased, while the content of the remaining bacteria (such as *Bacteroidetes*) gradually increased, and the abundance of *Methanosarcina* in archaea under different conditions was at a relatively high level [116]. Similarly, Zhang et al. studied the fermentation of FW under sweat sludge and microwave pretreatment, and the results showed that gas production after microwave pretreatment was improved. High-throughput sequencing analysis showed that the composition of the microbial community also changed significantly; *Bacteroides* dominated before pretreatment, and *Methanosphaera* and *Methanosarcina* became the main components of the microbial community after pretreatment [117]. High-throughput sequencing technology can help us better understand the relationship between the stability of the reaction process and the microbial community in the environment under different process technologies, and the technology provides a basis for efficiently regulating the reaction parameters and ensuring the smooth progress of the reaction.

**Table 3 ijerph-19-09519-t003:** Bacterial and archaea taxonomic composition at phylum/order and genus level in anaerobic digestion reaction determination by high-throughput sequencing.

Feedstocks	Scale	Dominant Phylum (Bacteria)	Dominant Genus (Bacteria)	Dominant Order (Archaea)	Dominant Genus (Archaea)	Sequencing Platform	Ref.
corn straw + chicken manure	1 L bottle	*Bacteroidetes Firmicutes Protecobacteria Chloroflexi Tenericutes*	Order: *Bacteroidales Clostridiales Xanthomonadales Lactobacillales Spirochaetales*	*Methanosarcinales Thermoplasmatales Methanobacteriales Methanomicrobiales*	*ND*	Illumina	[118]
Anaerobic sludge + food wastewater	50 L CSTR reactor	*Firmicutes Bacteroidetes Chloroflexi Actinobacteria Synergistetes*	Order: *Sphingobacteriales Bacillales Synergistales Thermotogales Clostridiales*	*Methanobacteriaceae Methanosaetaceae Methanosaetaceae Methanomicrobiaceae*	*Methanobacterium Methanosaeta Methanoculleus Methanolobus Methanosphaera*	Illumina	[119]
FW + Sludge	400 mL anaerobic bottles	*Chloroflexi Bacteroidetes Synergistetes* *Proteobacteria Firmicutes*	*Sutterella Treponema Phascolarctobacterium Bifidobacterium Bacteroides*	*Methanomicrobiales Thermoplasmatales Methanobacteriales Methanosarcinales*	*Methanosarcina,* *Methanoculleus Methanospirillum Methanobacterium Methanosaeta*	Illumina	[88]
Anaerobic sludge	118 mL reactor	*Actinobacteria Bacteroidetes Chloroflexi Firmicutes Spirochaetes*	*Coprococcus Mesotoga Cloacamonas Clostridium Treponema*	*Methanosacrinales Methanomicrobiales Thermotogae Methanobacteriales*	*Methanothrix Methanoculleus Methanolinea Methanosaeta Methanobacterium*	Illumina	[120]
FW + Seed sludge	three 6 L glass reactors	*Thermotogae Tenericutes Chloroflexi Bacteroidetes Firmicutes*	*Rikenellaceae Anaerolineaceae Clostridiales Gelria Barnesiella*	*Thermoplasmatales Methanosarcinales Methananomicrobiales Methanobacteriales*	*ND*	454	[116]
FW + Sludge	50 L CSTR reactor	*Firmicutes Bacteroidetes Nitrospirae* *Spirochaetes*	*Actinomyces Fastidiosipila Proteiniphilum Mobilitalea Aminobacterium*	*Methanobacteriales Methanomicrobiales Methanosarcinales*	*Methanosaeta Methanosarcina Methanobacterium Methanospirillim*	Illumina	[121]
FRW + DWW	24 L AnCMBR	*Bacteroidetes Fiemicutes Nitrospirae Proteobacteria* *Spirochaetes*	*ND*	*Methanobacterials Methanomicrobiales Methanosacrinales*	*ND*	Illumina	[122]

### 3.7. Combination of High-Throughput Sequencing Technology and Isotope Tracing Technology

In addition, the combination of high-throughput sequencing technology and isotope tracing technology has further enriched the research on anaerobic microbial communities (especially methanogens). The role of a specific population in the physiological and biochemical processes of fermentation can accurately be inferred by high-throughput sequencing technology. To investigate the effect of ammonia concentration on the methanogen composition and methanogenic pathways in the FW anaerobic digestion process, Jiang et al. used the radioactive isotope 14C, labeled acetic acid, to confirm that 68–75% of methane is syntrophic at high ammonia concentrations. In acetate oxidation coupled with a hydrogenotrophic methanogenesis pathway and low ammonia concentrations, the hydrogenotrophic pathway accounts for only 9–23% of methane production. In addition, using high-throughput sequencing technology based on the mcrA gene of methanogens for systematic community analysis, the diversity of methanogens in the samples will be particularly rich under different ammonia concentrations [123]. Because radioisotopes have certain risks, the tracking technology based on stable isotopes, also known as stable nucleic acid probe technology (DNA-SIP), is sparingly applied. This technology can combine the structural composition of microorganisms in complex environments with their physiological functions, and the combination with high-throughput sequencing technology effectively overcomes the bottleneck involved with this technology and significantly improves the speed and sensitivity of 13C-labeled DNA detection. This has special application in the study of anaerobic microorganisms [124]. For instance, Zou et al. studied the effect of pre-fermentation of ethanol on methanogenic flora and methanogenic pathways in FW at moderate temperature. Using 13C-labeled ethanol as a matrix, the analysis showed that 59.3% of the methane was obtained by acetic-acid-type methanogens, and the methane produced via the CO_2_ reduction pathway increased by 4.7%. Microbial community analysis by high-throughput sequencing showed that the relative abundances of *Clostridium* and *Methanobacterium* increased by 7.6% and 10.2%, respectively [125]. It can be seen that the use of isotope tracing technology can reveal the actual contribution rates of different methanogenic processes in the FW anaerobic digestion process, and high-throughput sequencing technology can be used to explore the changes in microbial community composition, further supporting this conclusion.

## 4. Perspective for the Study of Anaerobic Microorganisms

### 4.1. Integration of Multi-Omics for Understanding of the Microbial Community Activity during FWAD Process

Despite increasing studies on the FWAD reaction, the complex interactions between the microbial community taxa and FW resource and products are largely unknown. A multi-omics integration strategy is essential to study microbial community diversity, metabolic diversity, biogeochemical functions of microbial interactions during the AD of FW (Figure 3).

Metagenomics, extracting all of the DNA in environmental samples and constructing a gene library, can aid us in studying the entire microorganisms in the samples and further explore the composition, functional activity, and the relationship between the microbial community and the environment [126]. This method opens up a new approach to microbial research, allowing researchers to closely link genomic information to the environmental background and to discover new microorganisms through functional screening [127]. Using metagenomics to carry out relevant research on the anaerobic digestion process of FW can help us understand the linkage between the richness of microbial diversity and the reaction conditions. For example, Laura Rabelo Leite et al. used a metagenomic analysis to study the performance of organic loading rate (OLR) in processing FW. The results showed that the increase in OLR significantly affected the community structures of bacteria and archaea. Under high OLR conditions, *Methanosaeta* nearly disappeared. The methanogenesis was mainly attributed to the hydrogen trophic methanogens, revealing the relationship between the operating conditions affecting the production of methane and microorganisms [128]. Metatranscriptomics can identify the functional genes that are expressed by the microbial community under AD conditions. Therefore, it is possible to establish the AD functional profile of the associated community. Because microbial function plays a crucial role in the nutrient (C, N, and P) transformations in the AD of FW, the nutrient transformation gene abundance variations may be uncovered.

Metaproteins with an intrinsic metabolic function can relate microbial activities to the identity of defined organisms in FWAD microbial consortia, providing new insights into the role of microbial diversity in the biogeochemical cycles of FWAD.

The FWAD process produces amino acids, organic acids, carbohydrates, sugar alcohols, polysaccharides, proteins, nucleic acids, and lipids, which are key carbon and nitrogen sources for microorganism involvement in FWAD, thus shaping the structure and function of the microbial community. However, the function and dynamics of those products during FWAD are not clear.

The multi-omics in the related research on anaerobic reactions have shown an advantage in the analysis of the microbial community structure and function on different levels and the ecological relationships between different microbial communities during AD. For example, Zhu et al. revealed through metagenomics and transcriptomics that by using acetic acid as the sole organic carbon source and adding H_2_, the interspecies’ competition and symbiosis between the microbial communities were affected in the anaerobic reactor. The results showed that after adding H_2_, the same nutritional relationship between the hydrogen-producing bacteria and the methanogenic bacteria was formed, and the abundance of the two increased significantly. However, the existence of different affinity groups for H_2_ concentration among the methanogenic bacteria resulted in “*Methanoculleus*” and “*Methanothermobacter*” having a competitive relationship. In addition, the transcription information indicated that there was an exchange of amino acids and carbon sources between the bacteria and archaea, thus forming a symbiotic relationship [129].

### 4.2. Synthetic Biology for Manipulation of Functioning Microbial Consortia during FWAD Process

In addition, synthetic biology is an emerging research field that provides a new opportunity for the understanding and manipulation of the FWAD biological systems. Synthetic biology combines the disciplines of biology, engineering, informatics, chemistry, and physics to design and create new biomolecules, novel artificial pathways, and biological systems. With the development of metagenomics, multi-omics, machine learning, and gene edition, such as clustered regularly interspaced short palindromic repeats (CRISPR), synthetic biology has been applied in many aspects, such as biofuels. In FWAD, the metabolic diversity of the microbial consortia enables the interaction among the microbial community members and allows them to engage in syntrophic interactions. With regard to improving the productivities and yields of FWAD renewable energy, synthetic microbial consortium engineering and synthetic functioning cell can be applied. For synthetic microbial consortium engineering, mimicking and/or outperforming the functions of the FWAD wild-type microorganism that meets FWAD bioprocessing requirements are the key objectives. The minimal microbial consortia with low energy consumption are expected to perform the key FWAD metabolic function and diversity. For the synthetic functioning cell, it would be desirable to find and genetically edit the key cell, which is involved in the FWAD bioprocess based on the investigation of the structure and function of microbial consortia.

## 5. Conclusions

Anaerobic digestion is a promising FW treatment method due to its advantages of obtaining waste recycling without secondary pollution. However, due to the complex microbial consortia of FWAD, it is easy to break the reaction balance in the process of AD, eventually leading to the termination of the reaction. Various technologies focus on the manipulation of key parameters, such as pH, temperature, and the C/D ratio, which have been developed to maintain the stability of AD and ultimately achieve the increase in resource yield. The AD process is a microbially mediated reaction involving different microorganisms. The aim of these technologies is keeping the balance between the microbial community and the supplied FW (carbon and nitrogen source). Therefore, the understanding of the microbial community composition and diversity, metabolic diversity, biogeochemical functions of the microbial interactions during the AD of FW is essential for FWAD management. With the aid of fast-developing biotechnology, ecological research has revealed the vast number and diversity of microorganisms in the environment. The basis of FWAD is to study microbial community response to FW source and the related ecological change. The genomic diversity, evolutionary dynamics, and ecological processes of FWAD microorganisms are tightly linked to the rate and flux of the energy and biogeochemical cycles and global FW management. The integration of multi-omics helps us understand the microbial community activity during FWAD, which will help us to better manage FW resources. Synthetic biology is an emerging research field and shows great potential for the understanding and manipulation of biological systems. Synthetic microbial consortium engineering and synthetic functioning cell would promote the FWAD renewable energy production.

## Figures and Tables

**Figure 1 ijerph-19-09519-f001:**
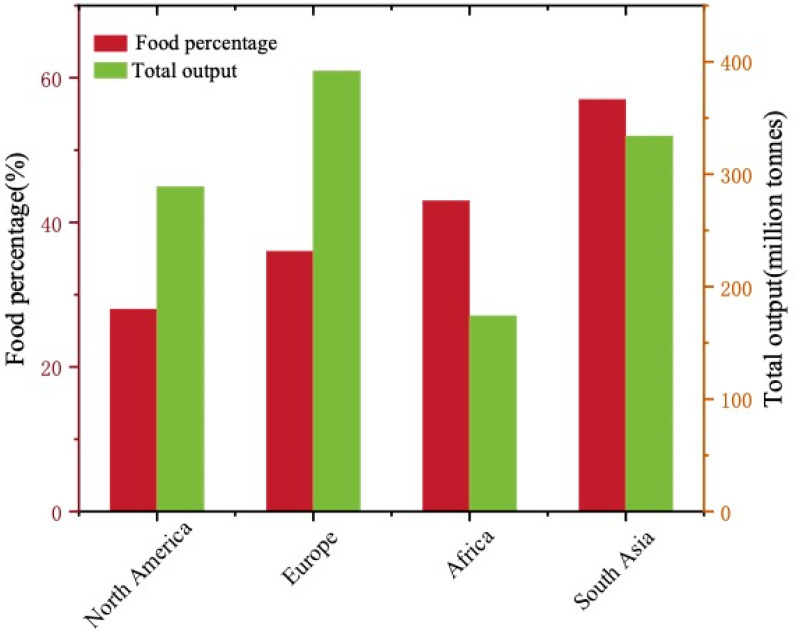
The total output of garbage and the proportion of food waste in different regions.

**Figure 2 ijerph-19-09519-f002:**
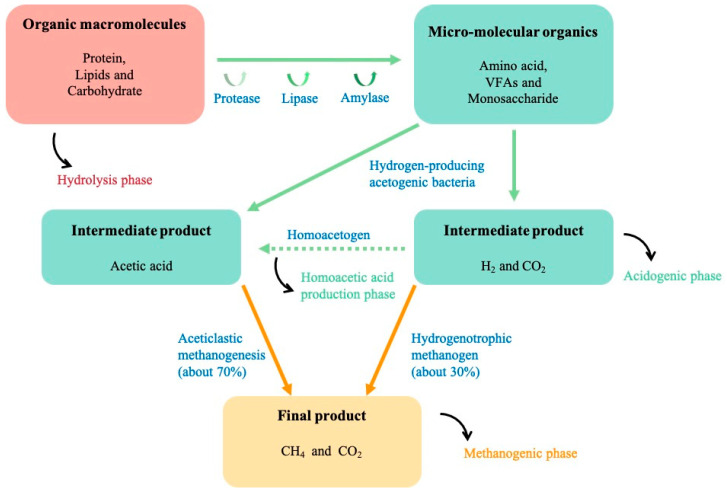
The process of anaerobic digestion of food waste (three-stage and four-stage systems).

**Figure 3 ijerph-19-09519-f003:**
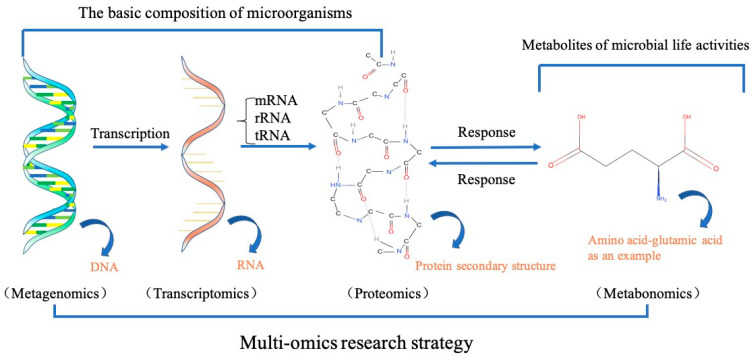
Sketch of multi-omics research strategy.

**Table 1 ijerph-19-09519-t001:** Different regulation methods of three important final products (H_2_, CH_4_, and organic acid) in FW anaerobic digestion process.

Reactor Volume	Final Products	Conditions	Results	Ref.
550 mL digesters	H_2_	Adjust the temperature to 55 °C	Achieve a maximum gas production of 82.47 mL/VS	[29]
500 mL digesters	Organic acid	Adjust the temperature to 37 °C	VFA maximum output is 34.4 g/L	[30]
500 mL serum bottles	CH_4_	Adjust the temperature to 35 °C	Gas production increased by 32% over 55 °C	[31]
635 mL fermenter	H_2_	Adjust the pH to 8.0	Maximum cumulative gas production is 1.3 L	[32]
4.5 L glass reactor	Organic acid	Adjust the pH to 6.0	Maximum acid production 40.89 g/L	[33]
500 mL experiment bottle	CH_4_	Adjust the pH to 8.0	7.57 times higher than pH uncontrolled	[34]
500 mL glass digesters	H_2_	Add ammonia soda	Maximum gas production is 145.4 mL H_2_/g-VS	[35]
430 ± 2 mL working volume	Organic acid	Add trace elements and activated carbon	A faster consumption of propionic acid	[36]
1 L batch reactors	CH_4_	Adjust the ammonia concentration to 0.5 g/L	Maximum gas production is 314.7 mL/g	[37]
4.5 L tank reactor	H_2_	Ultrasonic pretreatment	Increase in hydrogen production by 75%	[38]
1 L tank reactor	Organic acid	Ultrasonic pretreatment	VFAs increased by 27.2%	[39]
500 mL serum bottles	CH_4_	Alkali pretreatment	Maximum methane production rate is 6.63 mL/h	[40]
250 mL serum bottles	H_2_	Co-digestion with aged refuse and sewage sludge	Significantly increased hydrogen concentration by 26.6%	[41]
5 L reactor	Organic acid	Co-digestion with waste-activated sludge	SCFA maximum is 690.9 mg COD/g-VS	[42]
1 L reactor	CH_4_	Co-digestion with cow dung	Maximum gas production is 233 mL/g-VSS	[43]

## Data Availability

Not applicable.

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
