# Peer review of "Anaerobic Digestion of Food Waste and Its Microbial Consortia: A Historical Review and Future Perspectives"

_ijerph, 2022, doi:10.3390/ijerph19159519_

Round 1

Reviewer 1 Report

The manuscript is written well and will nicely contribute to the research field. Several comments and suggestions are provided below:

1. The title should be changed to something like “Anaerobic Digestion of Food Waste and its microbial analysis: A Historical…”

2. Introduction L79-L85: please also include the rationale to select/apply different molecular techniques in different studies on microbiome in FWAD

3.  In section 2, It would be nice to provide a statement to describe the strategy to select the final product to be H2, organic acids or CH4 and create a figure of FWAD process including selective conditions (e.g., optimal temp., pH and HRT) to obtain different target products, i.e., H2, organic acids, and CH4 (or modifying Figure2)

4. In Table 1:

- Table title, please leave out ‘Effect of’
- Column 1 ‘Raw materials’ can be removed, as all of them is FW which is mentioned already in the table title
- Column 2 ‘Scale’ can be changed to ‘Reactor volume’
- Column 3 ‘Substances’ should be changed to ‘Final products’

5. H2 production rate at temperatures above 35oC (e.g. 55 and 70oC) should also be mentioned (with references), even they are not for practical application. Does the rate still increase or reduce?

6. In Section 2.1.2. pH, please also provide optimal (low) pH range for H2 production that CH4 formation is not occurred

7. Line 179-182: Please provide an optimal ammonium concentration range applied to the system in general

8. At L355, please remove ‘(chemical oxygen demand)’ which is already provided in L276

9. Section 3.1. can be combined with the 1st paragraph of Section 3, so it can start with Section 3.1 Application of DGGE technology

10. It is highly suggested to include a table of Pros and Cons of different molecular techniques used to study microorganisms in the FWAD processes

Reviewer 2 Report

As a review article Points 1 and 2 (Introduction and The current state of AD of FW) must be renewed and actualized. Data from 2015 are not relevant today. The figures are instructive but not scientific at all, please remove them.

Although the topic of the article is very important, the real target of the publication is not clarified enough. 

Major revision is needed.

Reviewer 3 Report

This paper is of high quality and a useful review for those that are interested in food waste processing via anaerobic digestion, for a better and deep understanding of the process.

I have only a few minor remarks. They are as follows:

Line 75: ”The fermentation broth also produced in the AD process contains rich N 75 and K elements” - also phosphorus, used in fertiliser formulations. This should also be mentioned.

Section 2.3. - Maybe mention somewhere that AD produces biogas, that is a mixture of CH4, CO2 and other gases, but mainly methane.

Congratulations to the authors.
